# The Moderating Effect of Athletes' Personal Values on the Relationship between Coaches' Leadership Behaviors and the Personal and Social Skills of Young Basketball Players

Juan Facundo Corti [1] , María Julia Raimundi [2,3,4] , Ignacio Celsi [1,2] , Octavio Alvarez [5] and Isabel Castillo [5,*]

1 Research Institute, Faculty of Psychology, University of Buenos Aires, Buenos Aires C1052AAA, Argentina
2 National Scientific and Technical Research Council (CONICET), Buenos Aires C1425FQB, Argentina
3 Institute of Basic, Applied and Technological Psychology (IPSIBAT), National University of Mar del Plata, Mar del Plata B7603ETK, Argentina
4 Faculty of Psychology and Human Relations, Interamerican Open University (UAI), Buenos Aires C1147AAU, Argentina
5 Department of Social Psychology, Faculty of Psychology and Speech Therapy, University of Valencia, 46010 Valencia, Spain
* Correspondence: isabel.castillo@uv.es

**Abstract:** Transformational coaching has been shown to increase the personal and social skills of adolescent athletes. Nevertheless, the latter's dispositional characteristics, such us personal values, could have a moderating effect on this relationship. The main objective of this work was to examine perceptions of coaches' transformational behaviors and the modulation of athletes' personal values in their relationship with personal and social skills development. Adolescents ($n$ = 309) of both genders (81.9% male) aged 13–19 years ($M$ = 16.10; $SD$ = 1.70) from 16 different basketball clubs participated in the study. All participants completed a socio-demographic questionnaire, the Transformational Coaching Questionnaire, the Portrait Values Questionnaire—Revised, and the Youth Experiences Survey for Sport. Results showed that, controlling for age and gender, transformational behaviors that highlight individual differences within the team, especially through fostering autonomy and creative thinking (i.e., intellectual stimulation), create contexts in which their athletes can develop personal and social skills to their fullest. However, in players with high self-transcendence and low self-enhancement, the increase in intellectual stimulation was not associated with higher personal and social skills, but social focus transformational behaviors (i.e., idealized influence and inspirational motivation) were. This study contributes to the field of positive adolescent development by showing evidence of both the individual and the social focus of transformational leadership behaviors to maximize personal and social skills, depending on which values their athletes consider important.

**Keywords:** transformational leadership; values; youth; positive development; life skills; basketball





## 1. Introduction

Positive youth development is a broad research and practical approach that includes different theories and perspectives sharing a central point of view: children and adolescents' growth should be underpinned by a consideration of their individual characteristics and their social context in order to develop their potential to the fullest [1]. These potentialities can be developed through the acquisition of internal assets such as commitment to learning, positive values, social skills, and positive identity, among others [2]. Sport practice has been identified as a way to promote positive youth development by fostering life skills [3–5], defined as skills that are transferable from the sport context to other non-sport contexts [1,6]. Thus, skills such as empathy, tolerance, patience, sharing, and understanding and accepting the emotions of others, usually classified as personal and social skills [7], are considered essential assets for positive youth development [3].

Sport-based life skills development in young athletes can occur in two ways [4]. First, sports practice requires the development of a specific set of skills needed for competent sport performance, such as emotional control and goal setting, which may be transferable to non-sport contexts [8]. On the other hand, sport can promote a personal change in athletes' lives by generating dispositional changes in their sporting life [6]. However, none of those pathways is automatically transferable to everyday life. On the contrary, sport-based positive youth development is determined by a complex interaction of various factors, including the quality of relationships with parents, coaches, and peers [9]. In relation to the latter, there is a need for research to identify coaching behaviors that facilitate supportive environments and enhance the personal and interpersonal competences of young athletes, thereby contributing to their overall well-being [10].

From a social cognitive perspective, coaches serve as crucial role models in promoting cognitive, psychosocial, and behavioral development among their athletes [11,12]. Specifically, transformational coaches emphasize the personal development of athletes, which may affect their lives beyond the sport context [13].

Transformational leadership theory [14] proposes that transformational coaches exert their influence by sharing a vision of future team states that represent both a challenge for the athletes and a team improvement [11]. Coaches with a transformational leadership style move their athletes beyond immediate self-interest by providing a positive behavioral model for them to follow (idealized influence) and influencing their motivation by giving them meaningful and attractive challenges in everyday activities (inspirational motivation). Likewise, these leaders also encourage athletes to think in different ways when facing new and old challenges (intellectual stimulation) and treat each player as a valued team member, recognizing individual differences with a supportive leadership style (individualized consideration [15]). Coaches who display transformational behaviors change their followers' basic values, beliefs, and attitudes, seeking greater team and individual well-being [16]. On the other hand, this theory distinguishes between transformational and transactional (i.e., traditional) leadership styles. The latter implies stablishing a clear relation between what athletes should do and what they will get in return (contingent reward) and coaches' supervision, both before an athlete's mistake (management-by-exception active) and after (management-by-exception passive). Transformational leadership augmented the positive effects of transactional leadership, in what was called the "augmentation effect", as empirically shown across different contexts and cultures (for a review, see [11]). Transformational leaders may use transactional strategies when appropriate, but they also raise or expand individual needs, inducing a belief in transcending self-interest for the sake of the team [14].

Thus, transformational coaching is highly recommended to obtain benefits for athletes, as it was positively related to various desirable outcomes in positive youth development [11], such as prosocial behaviors [17], and has been shown to increase personal and social skills [18]. Vella et al. developed a transformational leadership-training program for coaches, and with a quasi-experimental design (with an experimental group and control group), they compared coaches' transformational leadership behaviors and personal and social skills variables of their teams' players [18]. Their results confirmed that athletes of teams with more transformational coaches had more positive experiences during their practice and developed more social and personal skills than athletes in less transformational coaching teams. Additionally, Newland et al. [13] studied coaches' specific characteristics to understand the relation between leadership behaviors and positive youth development. Results showed that coaches' transformational leadership behaviors significantly predicted positive youth experiences (i.e., competence, confidence, character, and caring) at both the individual and team levels.

Nevertheless, little attention has been paid to the dispositional characteristics of followers in the relation between leadership behaviors and positive youth development [19]. Some studies have found that the relationship between leadership behaviors and desired outcomes in athletes, such as extra effort, was moderated by athletes' level of narcissism [20]. Williams et al. [21] also found that trait anxiety and confidence levels affect the way that

athletes perceive their coaches' leadership behaviors. Among the dispositional variables that moderate the association between coaches' leadership style and athletes' personal and social skills, the latter's personal values could have explanatory potential, as values influence how people perceive, feel, and behave (for a review, see [22]).

Sports have been considered a good scenario to transmit and foster personal values [12,23], contributing to the positive development of young athletes. The refined theory of personal values [24] is a solid framework to understand values in sport and non-sport contexts [15,23,25,26]. In this framework, personal values are defined as desirable goals, stable across situations and time, that serve as guides for the conscious organization of emotions, attitudes, and behaviors [24]. Schwartz's refined value theory includes 19 value domains organized into two higher-order dimensions. One dimension opposes values that emphasize independence of judgment and action and favor change (i.e., openness to change) with those that emphasize preservation of traditions and protection of stability (i.e., conservation). The second dimension involves self-enhancement versus self-transcendence, contrasting values that emphasize the pursuit of personal success and dominance over others with values that emphasize acceptance of others as equals and concern for their well-being [24].

Basketball is a team sport that requires players to work together in a coordinated manner on a much smaller court than other sports, such as football. In addition, in games, there are moments of stoppage of the game where coaches can use feedback as a tool to build athlete confidence and help them to monitor progress [27]. Given these characteristics and the cultural importance of basketball in many countries, it has been chosen as a specific sport for the development of life skills programs, such as Personal and Social Responsibility [28]. This program proposes specific teaching and coaching strategies with a series of levels that attempt to maintain a balance between empowering athletes to make decisions for themselves and teaching them specific socially accepted values [29].

Focusing on the study of coach leadership behaviors and values, previous research has established that basketball coaches' personal values predict their levels of transformational behavior [15]. As far as we know, no study has examined associations between coaches' transformational leadership behaviors and their athletes' values. The present study aims to go a step further and analyze how athletes' personal values modulate their perception of such transformational behaviors, enabling or hindering greater personal and social skills development. To this end, the relationship between coaches' transformational leadership and the personal and social skills of their athletes will be assessed by achieving two objectives. First, this study will examine which specific transformational leadership behaviors best predict the development of athletes' personal and social skills. Second, the moderating role of personal values in this relationship will be analyzed, while controlling for age, gender, and number of seasons with their coach.

Accordingly, we hypothesize that (1) all four coaches' transformational behaviors predict the personal and social skills of their players, and (2) both self-transcendence and self-enhancement values significantly moderate the relationship between transformational behaviors and personal and social skills. However, (3) openness to change and conservation values are not expected to modulate this relationship. Finally, (4) control variables (i.e., age, gender, and seasons training with their coach) are associated with personal and social skills.

## 2. Materials and Methods

### 2.1. Study Design

An associative strategy was used with the implementation of a non-experimental cross-sectional predictive design [30].

### 2.2. Participants

Using Green's rule of thumb [31] for multiple regression analysis with a medium effect size and a total of 12 potential predictors, a minimum sample size of 150 participants was estimated. Participants were 309 basketball players (81.9% male) aged 13–19 years

(*M* = 16.10, *SD* = 1.70). They were recruited via convenience sampling from 16 different Argentinean clubs. All the institutions included in this study had at least one team per age category (i.e., U15, U17, and U19). Most athletes have been training with their team for more than two seasons (*M* = 2.38; *SD* = 1.40). On average, teams trained two times per week and competed in one game on the weekend.

*2.3. Measures*

The Transformational Coaching Questionnaire (TCQ [32]; Spanish version from [33]) was used to measure coaches' transformational behaviors. This questionnaire consists of 16 items that use a six-point Likert scale ranging from 1 (Not at all) to 6 (Frequently), to reflect players' perceptions about their coaches' transformational behavior: idealized influence (e.g., "Acts as a person that I look up to"), inspirational motivation (e.g., "Is enthusiastic about what I am capable of achieving"), intellectual stimulation (e.g., "Encourages me to look at issues from different sides"), and individualized consideration (e.g., "Shows that s/he cares about me"). High internal consistency reliability ($\alpha > 0.85$) has been reported for all subscales [33]. The internal consistency of all subscales was acceptable in the present study ($\alpha$ from 0.74 to 0.86, see Table 1).

Personal values were measured using the Spanish version of the 57-item Portrait Values Questionnaire—Revised (PVQ-R [24]), by asking players to indicate how similar they are to gender-matched individuals who are described in terms of their important values. The questionnaire starts with the stem, "How much is this person like you?", and responses ranged from 1 (not like me at all) to 6 (very much like me). An example of an item is: "It is important to him to have a good time". The refined model postulates 19 values, each measured with three marker items, which make up four major dimensions: self-transcendence, self-enhancement, openness to change, and conservation. The reliability of this instrument has already been demonstrated to be good ($\alpha > 0.81$ [15]). Cronbach's alphas in this study were also satisfactory ($\alpha$ from 0.79 to 0.87, see Table 1).

**Table 1.** Descriptive statistics, Cronbach's alpha, and gender differences.

| | | | Total | | Males | | Females | | |
|---|---|---|---|---|---|---|---|---|---|
| | $\alpha$ | Range | *M* | *SD* | *M* | *SD* | *M* | *SD* | *p* |
| Age | - | 13–19 | 16.17 | 1.71 | 16.23 | 1.70 | 15.86 | 1.75 | 0.14 |
| Seasons with their coach | - | 1–4 | 2.19 | 1.18 | 2.21 | 1.17 | 2.07 | 1.23 | 0.42 |
| Transformational Leadership | | | | | | | | | |
|    Idealized influence | 0.76 | 1–6 | 4.19 | 1.09 | 4.24 | 1.06 | 3.96 | 1.17 | 0.09 |
|    Inspirational motivation | 0.86 | 1–6 | 4.79 | 1.07 | 4.78 | 1.04 | 4.81 | 1.20 | 0.86 |
|    Intellectual stimulation | 0.74 | 1–6 | 4.25 | 1.03 | 4.26 | 1.00 | 4.21 | 1.16 | 0.09 |
|    Individualized consideration | 0.80 | 1–6 | 4.73 | 1.01 | 4.75 | 0.96 | 4.63 | 1.20 | 0.41 |
| Personal Values | | | | | | | | | |
|    Self-transcendence | 0.86 | 1–6 | 4.70 | 0.67 | 4.61 | 0.66 | 5.13 | 0.54 | 0.00 |
|    Self-enhancement | 0.79 | 1–6 | 3.52 | 0.86 | 3.61 | 0.84 | 3.12 | 0.85 | 0.00 |
|    Openness to change | 0.82 | 1–6 | 4.90 | 0.62 | 4.85 | 0.64 | 5.12 | 0.50 | 0.08 |
|    Conservation | 0.87 | 1–6 | 4.12 | 0.74 | 4.10 | 0.75 | 4.20 | 0.69 | 0.34 |
| Positive Youth Development | | | | | | | | | |
|    Personal and social skills | 0.61 | 1–6 | 5.10 | 0.64 | 5.07 | 0.66 | 5.26 | 0.54 | 0.04 |

Note. *p*-values were obtained with Student's *t*-tests.

The Youth Experiences Survey for Sport (YES-S [7]; short-form from [34]) was used to measure positive developmental experiences. The short form of this survey consists of 22 items that use a six-point Likert scale ranging from 1 (Not at all) to 6 (Yes, definitely) to reflect the players' experiences during the season. These items form five subscales: Personal and Social Skills, Cognitive Skills, Goal Setting, Initiative, and Negative Experiences. For

the present study, only the Personal and Social Skills subscale was considered (i.e., four statements: "I became better at sharing responsibility", "I learned that working together requires some compromising", "I learned to be patient with other group members", and "I learned how my emotions and attitude affect others in the group"), where higher scores indicate greater development. High internal consistency reliability ($\alpha > 0.82$) has been reported for all subscales [7]. In the present study, the internal consistency of the Personal and Social Skills subscale was marginal ($\alpha = 0.61$), with no improvement by eliminating some of its items (see Table 1).

### 2.4. Procedure

The research was conducted in accordance with the Declaration of Helsinki, and institutional approval was not necessary as no information was collected that could be sensitive to the participant. Participants completed the questionnaires in approximately 20 min, at a time conveniently scheduled around their team training sessions in the last few weeks of the season. They were asked to respond to questions regarding their current team and coach and were reminded that their participation was voluntarily and responses were confidential. To complete all questionnaires, participants used their mobile devices, which were connected to the research assistant's computer via Wi-Fi, where responses were automatically saved in a comma-separated values file. Prior to completing the questionnaires, written informed consent was obtained from all subjects involved in the study. Additionally, in accordance with current Argentine legislation, written parental consent was required for players under 16 years old.

### 2.5. Data Analysis

Data were transferred to R (v. 4.1.2) [35] for statistical analysis. Transformational behavior, personal values, and personal and social skills were computed as the mean score of each subscale item for each player. In moderation models, where one value dimension at a time was included, the scores for those values were centered on each participant, as suggested by Schwartz [36]. Student's *t*-test was performed to analyze differences across gender in all variables, and Pearson correlation coefficient was used to estimate the association between variables. On an exploratory basis, simple linear regression models were tested to predict personal and social skills with age, gender, seasons with their coach, all transformational behaviors, and all personal values as predictors. A multiple linear regression model with all four transformational behaviors was tested. As the residuals plot showed a marginal fit of that model, the Box–Cox transformation [37] and a gamma generalized linear model were tested, with no improvement. Therefore, it was decided to continue modeling linear regressions with the original variables. A moderation effect of personal values was added to the multiple linear model, and the effects of age, gender, and seasons with coach were controlled. Finally, the *leaps* algorithm [38] was used to select the best subset of terms to include in the final models, according to the Akaike information criterion (AIC) and mean squared error (MSE) estimated with leave-one-out cross-validation. Standardized and unstandardized coefficients, standard error, *F* values, *p*-values, and adjusted $R^2$ of final models are reported.

## 3. Results

Most players perceived their coaches as moderately high transformational leaders and reported higher levels of openness to change and self-transcendence values, compared to conservation and self-enhancement values. Athletes reported feeling personally and socially skilled. The differences between genders are shown in Table 1.

Student's *t*-test revealed significant gender differences for personal values and personal and social skills. Females gave more importance than males to self-transcendence (Cohen's $d = 0.82$) and reported higher personal and social skills (Cohen's $d = 0.32$), whereas males consider self-enhancement values more relevant than female (Cohen's $d = 0.57$).

Correlation analysis showed that personal and social skills are positively related to all four transformational behaviors and self-transcendence values, and negatively associated with self-enhancement values. Positive relationships were found between self-transcendence and inspirational motivation and intellectual stimulation. Conservation values were positively correlated with intellectual stimulation and individualized consideration. Finally, self-enhancement values were negatively associated with inspirational motivation, intellectual stimulation, and individualized consideration (see Table 2).

**Table 2.** Bivariate Pearson correlation matrix of the study variables.

| | 1 | 2 | 3 | 4 | 5 | 6 | 7 | 8 | 9 | 10 |
|---|---|---|---|---|---|---|---|---|---|---|
| 1. Age | - | | | | | | | | | |
| 2. Seasons with their coach | 0.12 * | - | | | | | | | | |
| 3. Idealized influence | −0.14 * | 0.16 ** | - | | | | | | | |
| 4. Inspirational motivation | −0.17 ** | 0.10 | 0.72 *** | - | | | | | | |
| 5. Intellectual stimulation | −0.21 *** | 0.05 | 0.72 *** | 0.62 *** | - | | | | | |
| 6. Individualized consideration | −0.14 * | 0.07 | 0.72 *** | 0.75 *** | 0.64 *** | - | | | | |
| 7. Self-transcendence | −0.10 | 0.05 | 0.05 | 0.14 * | 0.12 * | 0.09 | - | | | |
| 8. Self-enhancement | 0.21 *** | 0.00 | −0.09 | −0.21 *** | −0.23 *** | −0.18 ** | −0.64 *** | - | | |
| 9. Openness to change | 0.10 | 0.14 * | 0.03 | −0.01 | −0.01 | −0.07 | 0.06 | −0.03 | - | |
| 10. Conservation | −0.19 ** | −0.13 * | 0.02 | 0.09 | 0.12 * | 0.14 * | −0.21 *** | −0.42 *** | −0.68 *** | - |
| 11. Personal and social skills | 0.03 | −0.06 | 0.27 *** | 0.27 *** | 0.37 *** | 0.26 *** | 0.21 *** | −0.35 *** | 0.08 | 0.10 |

Note. * $p < 0.05$, ** $p < 0.01$, *** $p < 0.001$. Centered responses for each value dimension were used.

In simple linear regression models, age ($p = 0.61$), seasons with their coach ($p = 0.30$), openness to change ($p = 0.15$), and conservation ($p = 0.07$) could not predict significantly personal and social skills. Meanwhile, simple models were statistically significant when the predictors were gender ($F(1,307) = 4.27$; $p = 0.04$; $R^2 = 0.01$), idealized influence ($F(1,307) = 23.72$; $p < 0.001$; $R^2 = 0.07$), inspirational motivation ($F(1,307) = 24.01$; $p < 0.001$; $R^2 = 0.07$), intellectual stimulation ($F(1,307) = 47.71$; $p < 0.001$; $R^2 = 0.13$), individualized consideration ($F(1,307) = 21.69$; $p < 0.001$; $R^2 = 0.06$), self-transcendence ($F(1,307) = 14.57$; $p < 0.001$; $R^2 = 0.04$), and self-enhancement ($F(1,307) = 41.75$; $p < 0.001$; $R^2 = 0.12$).

Multiple linear regression model was performed with all four transformational behaviors, and only intellectual stimulation ($β = 0.21$; $p < 0.001$) was significant in this model ($F(4,304) = 12.16$; $p < 0.001$; $R^2 = 0.13$; AIC = 569.34; MSE = 0.37). Due to some structure in the residuals plot, the Box–Cox transformation (AIC = 1544.49; MSE = 8.67) and a Gamma generalized linear model (AIC = 602.98; MSE = 0.37) were tested, but neither enhanced the fit. The results of including personal values in the model are presented in Table 3, which summarizes the four linear models (one for each value) that combine the variables in such a way as to obtain the lowest AIC and MSE, according to the leaps algorithm.

All the best models predicting personal and social skills according to *leaps* algorithm included age and intellectual stimulation as predictors, while none included the number of seasons training with their coach. The proportion of the variance of personal and social skills explained by all models was acceptable ($R^2 = 0.17$; 0.24).

In the first model, the moderating effect of self-transcendence was significant for intellectual stimulation and idealized influence (see Figure 1a). Higher levels of self-transcendence predicted higher levels of personal and social skills for almost the whole range of intellectual stimulation, and high levels of idealized influence also boosted personal and social skills levels. However, when self-transcendence levels were low, higher levels of intellectual stimulation were associated with higher levels of personal and social skills, and idealized influence had no relevant effect. Both age and gender significantly predicted personal and social skills, with older participants and females having higher levels of personal and social skills.

**Table 3.** Linear regression models predicting personal and social skills, with transformational behaviors as predictors and moderating effects of each personal values, controlling for age and gender when included by leaps algorithm.

1. Self-transcendence as moderator. $F(5,303) = 16.58$; $p < 0.001$; $R^2 = 0.20$.

| Variable | B | St. Error | β | p |
|---|---|---|---|---|
| Intercept | 3.01 | 0.40 | - | 0.00 |
| Age | 0.05 | 0.02 | 0.13 | 0.01 |
| Gender (Male) | −1.20 | 0.09 | −0.12 | 0.03 |
| Idealized influence (II) | −0.03 | 0.05 | −0.06 | 0.52 |
| Intellectual stimulation (IS) | 0.37 | 0.06 | 0.57 | 0.00 |
| Self-transcendence (ST) | 1.30 | 0.38 | 0.80 | 0.00 |
| IS × ST | −0.45 | 0.11 | −1.26 | 0.00 |
| II × ST | 0.20 | 0.10 | 0.55 | 0.04 |

2. Self-enhancement as moderator. $F(4,304) = 24.91$; $p < 0.001$; $R^2 = 0.24$.

| Variable | B | St. Error | β | p |
|---|---|---|---|---|
| Intercept | 2.65 | 0.4 | - | 0.00 |
| Age | 0.07 | 0.02 | 0.17 | 0.00 |
| Inspirational motivation (IM) | −0.04 | 0.05 | −0.06 | 0.49 |
| Intellectual stimulation (IS) | 0.33 | 0.06 | 0.52 | 0.00 |
| Self-enhancement (SE) | −0.54 | 0.20 | −0.67 | 0.01 |
| IS × SE | 0.18 | 0.06 | 1.06 | 0.00 |
| IM × SE | −0.10 | 0.05 | −0.64 | 0.01 |

3. Openness to change as moderator. $F(8,300) = 9.05$; $p < 0.001$; $R^2 = 0.17$.

| Variable | B | St. Error | β | p |
|---|---|---|---|---|
| Intercept | 3.30 | 0.45 | - | 0.00 |
| Age | 0.05 | 0.02 | 0.14 | 0.01 |
| Gender (Male) | −0.22 | 0.09 | −0.13 | 0.01 |
| Inspirational motivation (IM) | −0.14 | 0.08 | −0.23 | 0.06 |
| Intellectual stimulation (IS) | 0.21 | 0.04 | 0.34 | 0.00 |
| Individualized consideration (IC) | 0.18 | 0.08 | 0.29 | 0.03 |
| Openness to change (OC) | −0.10 | 0.36 | −0.07 | 0.78 |
| IM × OC | 0.26 | 0.09 | 0.89 | 0.00 |
| IC × OC | −0.23 | 0.10 | −0.75 | 0.02 |

4. Conservation as moderator. $F(6,302) = 11.30$; $p < 0.001$; $R^2 = 0.17$.

| Variable | B | St. Error | β | p |
|---|---|---|---|---|
| Intercept | 3.16 | 0.42 | - | 0.00 |
| Age | 0.06 | 0.02 | 0.15 | 0.01 |
| Gender (Male) | −0.26 | 0.09 | −0.15 | 0.00 |
| Intellectual stimulation (IS) | 0.21 | 0.04 | 0.34 | 0.00 |
| Individualized consideration (IC) | 0.09 | 0.05 | 0.13 | 0.08 |
| Conservation (C) | −0.61 | 0.43 | −0.35 | 0.15 |
| IC × C | 0.16 | 0.09 | 0.44 | 0.07 |

Note: Centered responses for each value dimension were used.

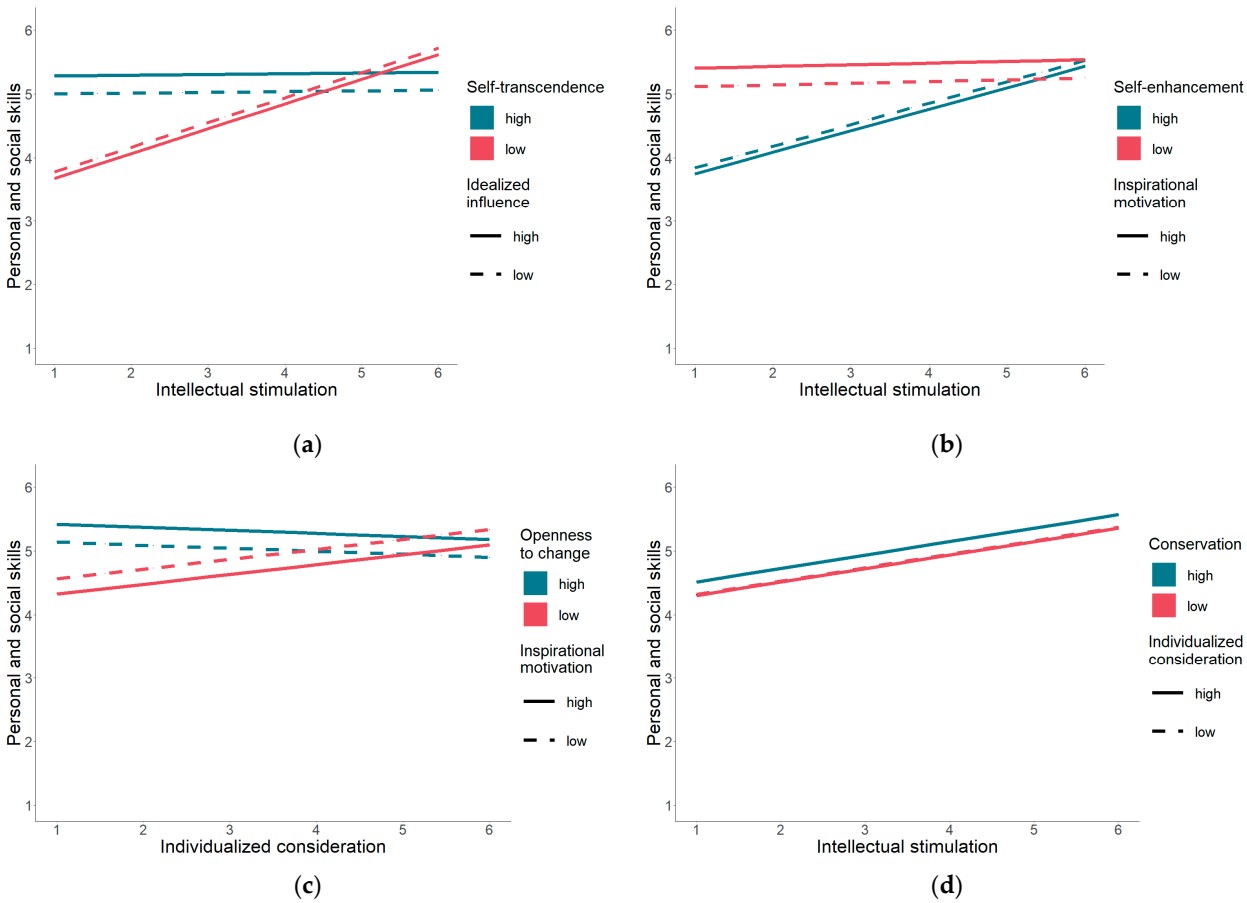

**Figure 1.** Models' predictions of personal and social skills according to the most relevant interactions in each model: (**a**) self-transcendence as a moderator; (**b**) self-enhancement as a moderator; (**c**) openness to change as a moderator; (**d**) conservation as a moderator. Quantiles 0.15 and 0.85 are considered low and high levels of each variable. In all models' predictions, other numeric variables were fixed at their mean, and gender at "male", when included.

The second model showed a significant moderating effect of self-enhancement on inspirational motivation and intellectual stimulation (see Figure 1b). Complementary to the model with self-transcendence, lower levels of self-enhancement predicted higher levels of personal and social skills for all intellectual stimulation levels. In this scenario, higher levels of inspirational motivation were associated with higher personal and social skills. On the contrary, when reported self-enhancement was low, inspirational motivation had no effect, and higher levels of intellectual stimulation predicted higher personal and social skills. In this model, age was a significant predictor of personal and social skills.

Openness to change interacted significantly with inspirational motivation and individualized consideration in predicting personal and social skills. Figure 1c shows how this interaction unfolds: higher levels of inspirational motivation predicted higher levels of personal and social skills when openness to change was low, and the opposite occurred when this value was high. Similarly, individualized consideration tended to be positively associated with personal and social skills when levels of openness to change were low, but it was negative associated when athletes reported high levels of openness to change. Intellectual stimulation, age, and gender also significantly predicted personal and social skills, with openness to change having no moderation effect.

Finally, the model that included conservation values as a moderator had no significant interactions. The main effect of conservation was also not significant. In this model, only age, gender, and intellectual stimulation significantly predicted personal and social skills values.

## 4. Discussion

The main goals of this study were twofold. First, to analyze which transformational coach behaviors best predict the development of personal and social skills in their players. Second, the moderating role of the four dimensions of personal values (i.e., self-transcendence, self-enhancement, openness to change, and conservation) in the aforementioned relationships was assessed.

In terms of our first hypothesis, the findings from single regression models demonstrate that all four transformational behaviors significantly predicted personal and social skills. However, regarding coaches' transformational behaviors that best predicted the personal and social skills of their players, intellectual stimulation was the only one included in all models. It was also the behavior with the greatest effect on the development of this set of skills. Previous studies in both Argentinian athletes [39] and other populations [40] established that when coaches ask their athletes what they think and encourage them to think of creative solutions, they are fostering their engagement in the activity. If players are engaged, they become involved with both the task and the team [41], further developing their personal and social skills. Tse and Chiu [42] deepen the conceptualization proposed by Wu et al. [43] in which they discriminate between the components of transformational leadership with individual focus (i.e., intellectual stimulation and individualized consideration) and those with social focus (i.e., idealized influence and inspirational motivation). When leaders adopt individually focused transformational behaviors, they highlight the individuality of each follower, recognizing them as distinct members in terms of their thoughts, emotions, and behaviors. In contrast, when transformational leaders adopt a social focus, they seek to influence the group as a whole, trying to generate shared values and a common ground, emphasizing the commonalities among members [42]. As operationalized in the YES-S, personal and social skills require recognizing differences among team members: learning that working together implies compromising, being patient with peers, and understanding that personal attitudes may affect others. All these skills involve accepting that one's teammates have different interests and needs and that every player must recognize individual differences in others in order to participate in their team. Although it may seem counter-intuitive, this is the way to really develop social skills, because if all members of the group were the same, there would not adapt to each other's needs, but would only attend to their own. With that in mind, it makes sense that the transformational leadership behavior that best predicted the personal and social skills levels was one with an individual focus (i.e., intellectual stimulation). In the context of team sports, such us basketball, when coaches value the perspective of each athlete and stimulate their creative thinking, they could set in motion a challenge-resolution mechanism in their players that necessarily includes the perspective of teammates, thus enhancing the consideration of other perspectives and the development of personal and social skills that are valuable in other life domains [28].

Lawrason et al. [44] described how transformational coaches may promote interdependence perceptions (i.e., athletes who understand their own success as a result of team members' success) on their teams by encouraging intellectual growth. In this sense, previous literature has highlighted how outcome interdependence is linked to positive relationships between players and promotes teamwork and social skills [45].

Similar conclusions can be drawn regarding the second hypothesis when analyzing the interactions in the models. Self-transcendence significantly moderates the relationship between transformational behaviors of intellectual stimulation and idealized influence and personal and social skills. Naturally, if caring for those close to them and being tolerant of those who are different (i.e., self-transcendence values) are important for athletes, then they will also show higher levels of personal and social skills. In these situations, the effect of coaches' intellectual stimulation may be minimal. However, having the coach talk about his or her personal values and act as a role model (i.e., idealized influence) generates a small increase in personal and social skills levels, functioning as a booster of the effect of self-transcendence on personal and social skills. In contrast, when self-transcendence

values are low, the interaction effect of idealized influence is null, but it has a large impact on personal and social skills to intellectually stimulate players. This result shows the importance of transformational coaching in the positive development of young athletes, especially when the adolescents' dispositional characteristics may hinder positive personal and social development.

When considering the model with self-enhancement as moderator, complementary conclusions can be drawn. When self-enhancement values are low, stimulating athletes intellectually does not generate great changes. However, the coach can boost personal and social skills in their athletes by providing support and attractive challenges (i.e., inspirational motivation). In contrast, when players focus on personal achievement and power over others, intellectual stimulation does enhance the development of personal and social skills, while inspirational motivation does not increase the level of these skills. Again, when the personal values prioritized by the athletes are detrimental to the development of personal and social skills (i.e., low levels of self-transcendence and high levels of self-enhancement), individual focused transformational behaviors are the ones that maximize the development of these skills. Conversely, when players exhibit high levels of self-transcendence and low levels of self-enhancement, coaches should focus on socially oriented transformational behaviors. This correspondence between the self-transcendence model and the self-enhancement model also provides evidence for the personal value structure proposed by Schwartz et al. [24], since in both the theoretical model and the moderation effect in this study these values are opposed to each other.

Contrary to our third hypothesis, openness to change modulates the effect of individualized consideration. When athletes do not particularly prioritize independence of thought and action, stimulation, or hedonism, their coach's consideration of their individual thoughts, interests, and needs promotes the development of personal and social skills. Conversely, when openness to change values are high, individualized consideration may inhibit the development of personal and social skills. When players seek their own independence and pleasure, emphasizing their individual needs may focus too much on individualities and be detrimental to the development of personal and social skills. Therefore, knowing the prevailing values of the team and each individual player can be crucial to adopting appropriate leadership behaviors that maximize positive youth development through sport. Arthur et al. [20] concluded similarly when evaluating the moderating role of athletes' narcissism in the relationship between transformational behavior and leader-inspired extra effort: considering followers' dispositional characteristics is crucial for assessing the impact of the leader's behavior on their athletes.

As previously established, coaches' values are related to transformational leadership behaviors [15], and the latter were associated with higher personal and social skills [18]. Moreover, López-Mora et al. [46] found that specific sport values of coaches may foster or hinder prosocial reasoning in their players. However, the results of the present study show that athletes' personal values may not only affect their perception of their coach's leadership style, but also moderate the coach's effect on the development of personal and social skills. Apart from that, previous research on basketball players' personal values [23] revealed the importance of this variable when focusing on positive youth development through this sport. Self-transcendence values were positively associated with autonomous motivation, wellbeing, and future intention to practice, and self-promoting values were negatively associated with these desirable outcomes [23]. Results found in our study suggest another reason for coaches to know their athletes' values profiles and the ones they are promoting in their training sessions and competences. As the different transformational leadership behaviors have different effects depending on the value profile of the athletes, knowing these values will allow coaches to create an optimal sport environment that facilitates the development of life skills in young people.

In the best-fitting model with conservation as a moderator, no significant effect of this type of value or its interaction with transformational leadership behaviors was found, as previously hypothesized. This result is consistent with other studies that show that the

value of conservation is not related to motivation or practice intention in young basketball players [23]. It would seem that values linked to maintaining traditions, feeling personal and socially secure, and complying with the rules, besides decreasing in importance at this life stage [47], are not linked to positive development.

Hypothesis four was partially supported. Both age and gender were significant predictors in all models, except for gender in the model that included the moderation of self-enhancement. As the positive youth development literature has established [48,49], the development of these skills occurs naturally in supportive contexts. The role of adults is primarily to support these trajectories so the adolescents can maximize their positive skills. Therefore, it makes sense that the older the age, the greater the development of personal and social skills, even with a low effect as found in the present study. In addition, males perceive themselves as significantly less developed in personal and social skills. In this vein, previous research in Argentinian young athletes [50] and other adolescent populations [51] has found that character strengths with social focus, like kindness, are more developed in girls and women. Likewise, women exhibited higher levels of moral [52] and prosocial [46] reasoning compared to men.

Contrary to expectations, none of the best-fitting models included the number of seasons in which athletes had trained with their coach as a significant predictor. This could have been due to the change of coach that occurs in adolescent sport when the athlete changes age category, given that a different coach usually leads each category. Thus, athletes change coaches every two years, which limits the effect that the duration of the coach–player relationship could have on the development of personal and social skills.

## 5. Conclusions and Limitations

The findings from this study extend previous results on the moderating role of athletes' personal traits in the association between their positive development and their coaches' transformational behavior. Coaches who adopt transformational behaviors that highlight individual differences within the team, especially through fostering autonomy and creative thinking, are creating contexts in which their athletes can develop personal and social skills to their fullest. However, in players with high self-transcendence and low self-enhancement, there seems to be a ceiling effect in which the increase in intellectual stimulation is not associated with higher personal and social skills. Furthermore, when independence of action and thought, stimulation, and hedonism are highly valued by athletes (i.e., high openness to change), being individually considered may hinder the development of personal and social skills, and focus should be set at socially orientated transformational behaviors (i.e., idealized influence and inspirational motivation). In sum, when considering the positive development of adolescents through sport, both the individual focus and the social focus of transformational leadership behaviors are necessary to maximize personal and social skills, depending on which values their athletes consider important.

The strengths of this study include gaining insight into the specific transformational behaviors that favor personal and social skills development, and how athletes' personal values moderate that association. Furthermore, no other studies have been found on the prediction of personal and social skills by coaches' transformational behaviors with athletes' personal values as moderators.

Nonetheless, some limitations need to be considered when interpreting the results. First, causal relationships cannot be established based on presented data due to the cross-sectional design. Therefore, future research should adopt experimental and longitudinal designs to strengthen the conclusions of this study. Second, although the sample size was double the estimated minimum, effect sizes in all models were smaller than expected. Therefore, the number of participants included in the study could be suboptimal. Third, the sample was unbalanced in terms of gender (i.e., females were under-represented), and only basketball players were considered. In Argentina, there are many more boys and men playing basketball than girls and women, and so this scenario is representative of the reality of this country. Although the effect of gender and age was controlled for inferential

analysis, future studies could address the different life stages of boys and girls and how these personal and contextual variables interact in the development of personal and social skills. In addition, further research is needed to extend the generalizability of these findings to other sports. Moreover, the data were collected in December 2021, a few months after sports activity resumed following the COVID-19 pandemic lockdown, which in Argentina included a year without training or sporting competitions. Therefore, this could interfere with the results found in this study. Finally, common method variance may have inflated the relationships hypothesized in this study due to the use of self-report measures, which are very common in sport psychology research. Future studies would benefit from the use of objective variables, such as observational methods to evaluate coaches' behaviors.

Despite these limitations, the results presented in this study shed light on the association between coaches' specific transformational leadership behaviors and their players' personal and social skills, while considering athletes' personal values as moderators of that relationship. Thus, using a sample of adolescent basketball players, previous research suggesting that followers' characteristics must be considered in the study of desired outcomes associated with leadership style in the sport context were extended.

## 6. Practical Implications

Based on the results of this study, it would be desirable to perform coach development programs that would enable coaches to recognize their own leadership strategies and their impact on youth development through sports. These workshops could promote coaches' self-awareness through reflection and discussion. In addition, these interventions could allow them to share common experiences and creative problem-solving in implementing transformational strategies through collaboration with other colleagues.

Moreover, coaches' programs developed through sport psychology professionals should aim to develop their ability to identify dominant values among their athletes in order to adjust their transformational behaviors effectively. Another objective could be to learn effective strategies to establish team values that all athletes could relate to, which could facilitate the task of coaches. Therefore, training programs for coaches should also consider the development of these types of skills.

**Author Contributions:** Conceptualization, J.F.C., I.C. (Ignacio Celsi) and O.A.; methodology, J.F.C. and I.C. (Isabel Castillo); software, J.F.C.; validation, I.C. (Isabel Castillo); formal analysis, J.F.C. and I.C. (Isabel Castillo); investigation, J.F.C., I.C. (Ignacio Celsi) and M.J.R.; resources, I.C. (Ignacio Celsi), M.J.R. and O.A.; data curation, J.F.C.; writing—original draft preparation, J.F.C. and O.A.; writing—review and editing, I.C. (Ignacio Celsi), M.J.R., O.A. and I.C. (Isabel Castillo); visualization, J.F.C.; supervision, M.J.R. and I.C. (Isabel Castillo); project administration, J.F.C. and M.J.R.; funding acquisition, J.F.C., M.J.R., I.C. (Ignacio Celsi), O.A. and I.C. (Isabel Castillo). All authors have read and agreed to the published version of the manuscript.

**Funding:** This research was funded by University of Buenos Aires, grant number REREC-2020-1245-E-UBA-REC, and a mobility grant under a framework agreement between the University of Valencia and the University of Buenos Aires for professors I. Castillo and O. Alvarez in 2022.

**Institutional Review Board Statement:** The study was conducted in accordance with the Declaration of Helsinki and approved by the Responsible Conduct in Research Committee of the Faculty of Psychology at University of Buenos Aires (Ref.: UBA17-05-21).

**Informed Consent Statement:** Written informed consent was obtained from all subjects involved in the study. Additionally, in accordance with current Argentine legislation, written parental consent was required for players under 16 years old.

**Data Availability Statement:** Publicly available datasets were analyzed in this study. This data can be found here: http://hdl.handle.net/11336/188096 (accessed on 1 October 2022).

**Acknowledgments:** We want to thank the athletes, coaches, parents, and authorities of the institutions that participated in this study: Club 17 de Agosto, Club Ateneo Popular Versailles, Club Banco Nación, Club Italiano, Club Pinocho, Club Tres de Febrero Club Unión Florida, Deportivo San Andrés, Gimnasia y Esgrima de Villa del Parque, Institución Sarmiento, and Sportivo Villa Ballester. In addition, we want to thank members of EIIPD (Equipo de Investigación e Innovación en Psicología del Deporte) for data collection, analysis, and ideas for this work.

**Conflicts of Interest:** The authors declare no conflict of interest.

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
