# Peer review of "The Moderating Effect of Athletes’ Personal Values on the Relationship between Coaches’ Leadership Behaviors and the Personal and Social Skills of Young Basketball Players"

_sustainability, doi:10.3390/su15054554_

Round 1
Reviewer 1 Report
Dear authors, Thank you very much for this interesting article, I think it can have a great impact and be suitable for this journal. However, I think you should take into account a number of considerations:
INTRODUCTION
- I consider it is very well written and developed. Perhaps, it would include some aspects related to the specific sport to be treated (basketball). What's more, there are studies that use the Personal and Social Responsibility Model (used to promote values ​​through sports) and its origins with Hellison were precisely in basketbal
METHOD: very well written, but would include study design
The results are very clarifying and the discussion too. Although, I think we should include the number of participants as a limitation. (in predictive studies it may be somewhat reduced). On the other hand, did you consider doing an inferential analysis by gender//age?
Maybe you could include some practical applications about the possibility of conducting intervention studies to improve the democratic leadership of coaches
For the rest, congratulations
Author Response
Responses to Reviewer 1
Dear authors,
Thank you very much for this interesting article, I think it can have a great impact and be suitable for this journal. However, I think you should take into account a number of considerations:
Response: We would like to thank Reviewer 1 for the positive comments on our manuscript. Below we show how we addressed each suggestion/comment made by the Reviewer.
INTRODUCTION
- I consider it is very well written and developed. Perhaps, it would include some aspects related to the specific sport to be treated (basketball). What's more, there are studies that use the Personal and Social Responsibility Model (used to promote values ​​through sports) and its origins with Hellison were precisely in basketball
Response: We thank the reviewer for this suggestion. We have included some aspects related to basketball, and its characteristics as a specific sport, as well as two studies that use Hellison's Personal and Social Responsibility Model, as suggested. (see page 3; lines 124-133).
METHOD: very well written, but would include study design
Response: We have included the study design (see page 4; lines 154-156).
The results are very clarifying and the discussion too. Although, I think we should include the number of participants as a limitation. (in predictive studies it may be somewhat reduced). On the other hand, did you consider doing an inferential analysis by gender//age?
Response: We thank the reviewer for this suggestion. We included the number of participants as a limitation (see page 12; lines 473-475). Respect to inferential analysis by gender/age, a future direction of the study considering this suggestion was included, given that the present study controlled for the effect of age and gender but did not perform analyses considering the potential differential effects on the life stages of male and female players (see page 12; lines 479-482).
Maybe you could include some practical applications about the possibility of conducting intervention studies to improve the democratic leadership of coaches
Response: Following the suggestion of the reviewer we have included some practical applications (see page 13; lines 498-509).
For the rest, congratulations
Response: Thank you again for your positive comments on our manuscript.
Reviewer 2 Report
First of all, I wanted to congratulate the authors for this work and thank the editorial team for allowing me to review it.
Regarding the manuscript, some modifications are necessary before its consideration for publication:
In the abstract it would be desirable to include some background information.
Introduction
Line 34: Positive youth development is a very generic term. They should try to be more specific.
You explain what a transformational coach is, but there are other types of coaches? How do they differ? What are the benefits and drawbacks of each type?
Line 73: "These authors", please specify the name of the authors since when citing in Vancouver format you do not know who you are talking about.
I would eliminate this type of indications (e.g., [15]), leaving only the reference number. In all the manuscript.
Please include hypothesis and discuss it according to the results.
Method
Missing sample size calculation.
It would be appropriate to include the study design.
Line 175: "to R (v. 4.1.2)". Information about the software and the country of the software is missing.
Results
They are well expressed and understandable.
Discussion
The discussion is not misguided and the arguments are logical, but it would be necessary to include more previous references to support these results and it would not seem to be a mere invention of the authors.
In addition, some sentences are excessively long and make reading difficult.
Conclusions and limitations
The conclusions and limitations seem to me to correctly represent the study carried out.
Author Response
Responses to Reviewer 2
First of all, I wanted to congratulate the authors for this work and thank the editorial team for allowing me to review it.
Response: We would like to thank Reviewer 2 for the positive comments on our manuscript. Below we show how we addressed each suggestion/comment made by the Reviewer.
Regarding the manuscript, some modifications are necessary before its consideration for publication:
In the abstract it would be desirable to include some background information.
Response: Following the suggestion of the reviewer we have included some background information in the abstract section (see page 1; lines 15-17).
Introduction
Line 34: Positive youth development is a very generic term. They should try to be more specific.
Response: Following the suggestion of the reviewer we have developed the term positive youth development (see page 1; lines 37-41).
You explain what a transformational coach is, but there are other types of coaches? How do they differ? What are the benefits and drawbacks of each type?
Response: We have conducted changes in the introduction section to clarify other types of coaches following the framework used in this study and the benefits of using each of them (see page 2; lines 75-86).
Line 73: "These authors", please specify the name of the authors since when citing in Vancouver format you do not know who you are talking about.
Response: Following the suggestion of the reviewer we have modified the expression used in previous line 73 trying to make the text more understandable (page 2; line 89).
I would eliminate this type of indications (e.g., [15]), leaving only the reference number. In all the manuscript.
Response: Following the suggestion of the reviewer we have deleted the expression e.g. in all the manuscript leaving only the reference number (lines 44, 62, 112, 114, 135, 410, 429, 435, 436).
Please include hypothesis and discuss it according to the results.
Response: Following the suggestion of the reviewer we have included the hypothesis and discussed it according to the results (see pages 3,10-12; lines 146-152, 320-322, 358, 389, 422, 427).
Method
Missing sample size calculation.
It would be appropriate to include the study design.
Line 175: "to R (v. 4.1.2)". Information about the software and the country of the software is missing.
Response: We have included information about missing sample size calculation (see page 4; lines 158-160), the study design (see page 4; lines 154-156) and the missing information about the software used in the study (see page 5, 15; lines 214, 612-613).
Results
They are well expressed and understandable.
Response: Thank you for your positive comment.
Discussion
The discussion is not misguided and the arguments are logical, but it would be necessary to include more previous references to support these results and it would not seem to be a mere invention of the authors.
In addition, some sentences are excessively long and make reading difficult.
Response: Following the suggestion of the reviewer we have included more references in the discussion section to support our findings (see pages 10,15; lines 325-326, 327, 352; 597-598, 619-628). Furthermore, we have also tried to reduce the length of some sentences to make them easier to read (see pages 10-11; lines 338-343, 361-364, 374-377, 409-414).
Conclusions and limitations
The conclusions and limitations seem to me to correctly represent the study carried out.
Response: Thank you again for your positive comments on our manuscript.
Round 2
Reviewer 2 Report
The authors have responded to all suggestions and the article is ready for publication.